# Proof-of-Concept: Antisense Oligonucleotide Mediated Skipping of Fibrillin-1 Exon 52

**DOI:** 10.3390/ijms22073479

**Published:** 2021-03-27

**Authors:** Jessica M. Cale, Kane Greer, Sue Fletcher, Steve D. Wilton

**Affiliations:** 1Centre for Molecular Medicine and Innovative Therapeutics, Health Futures Institute, Murdoch University, Murdoch, WA 6150, Australia; jessica.cale@murdoch.edu.au (J.M.C.); K.Greer@murdoch.edu.au (K.G.); S.Fletcher@murdoch.edu.au (S.F.); 2Centre for Neuromuscular and Neurological Disorders, Perron Institute for Neurological and Translational Science, The University of Western Australia, Nedlands, WA 6009, Australia; 3PYC Therapeutics, Nedlands, WA 6009, Australia

**Keywords:** Marfan syndrome, fibrillin-1, antisense oligonucleotides, exon skipping, splice-switching

## Abstract

Marfan syndrome is one of the most common dominantly inherited connective tissue disorders, affecting 2–3 in 10,000 individuals, and is caused by one of over 2800 unique *FBN1* mutations. Mutations in *FBN1* result in reduced fibrillin-1 expression, or the production of two different fibrillin-1 monomers unable to interact to form functional microfibrils. Here, we describe in vitro evaluation of antisense oligonucleotides designed to mediate exclusion of *FBN1* exon 52 during pre-mRNA splicing to restore monomer homology. Antisense oligonucleotide sequences were screened in healthy control fibroblasts. The most effective sequence was synthesised as a phosphorodiamidate morpholino oligomer, a chemistry shown to be safe and effective clinically. We show that exon 52 can be excluded in up to 100% of *FBN1* transcripts in healthy control fibroblasts transfected with PMO52. Immunofluorescent staining revealed the loss of fibrillin 1 fibres with ~50% skipping and the subsequent re-appearance of fibres with >80% skipping. However, the effect of exon skipping on the function of the induced fibrillin-1 isoform remains to be explored. Therefore, these findings demonstrate proof-of-concept that exclusion of an exon from *FBN1* pre-mRNA can result in internally truncated but identical monomers capable of forming fibres and lay a foundation for further investigation to determine the effect of exon skipping on fibrillin-1 function.

## 1. Introduction

Marfan syndrome (MFS, MIM 154700) is one of the most common dominantly inherited connective tissue diseases, affecting an estimated 2–3 in 10,000 individuals [1,2], in a family of disorders called the type-1 fibrillinopathies [3]. Marfan syndrome is characterised by extreme height with disproportionate limb and digit length in comparison to the torso, coupled with a myriad of other skeletal, ocular, skin and cardiovascular abnormalities [4]. However, it is the progressive growth of the aorta often eventuating into aortic dissection and rupture that is the most common cause of death [5].

Marfan syndrome was linked in the early 1990s to mutations in the, then recently discovered [6], fibrillin-1 gene (*FBN1*) [7,8]. Since then, over 2800 disease-causing mutations have been reported [9]. Fibrillin-1 encodes a large 350 kDa glycoprotein of the same name that is secreted from the cell and deposited into the extracellular matrix (ECM) [6]. In a healthy individual, fibrillin-1 monomers aggregate into multimer units within the first few hours after secretion [10]. Fibrillin-1 multimers form the backbone of microfibrils [6] that are essential in the majority of connective tissues and to which many microfibril associated proteins bind [11]. It is in the microfibril form that fibrillin-1 exerts its structural and regulatory roles, providing a backbone for microfibrils [12], maintaining the stability of elastic fibres [13], and regulating the bioavailability of signalling proteins such as transforming growth factor-beta (TGF-β) [14,15].

In a Marfan syndrome patient, the disease-causing *FBN1* mutation results in a lack of functional microfibrils, in turn leading to instability of the ECM that is further compounded by the dysregulation of TGF-β [12,16]. An increase in bioavailable TGF-β activates a signalling cascade resulting in, among other outcomes, increased expression of matrix metalloproteinase [15] that degrade fibrillin-1 and other matrix proteins leading to further destabilisation of the ECM [15,17]. The initial loss of functional microfibrils is theorised to depend on the type of mutation. In general, missense mutations, which do not affect a conserved cysteine, as well as splicing mutations are thought to exert dominant negative effects. Such mutations result in the production of a dominant aberrant monomer that disrupts the assembly of the wild-type protein into microfibrils [18]. In contrast, many nonsense and frameshifting mutations are associated with haploinsufficiency [19,20]. This haploinsufficiency is the result of transcript instability that leads to degradation and thus reduced fibrillin-1 expression [21]. A small subgroup of nonsense and frameshift mutations that affect the C-terminal region can produce stable transcripts that are translated into protein rather than being degraded [22,23]. Such mutations have been associated with intracellular retention of fibrillin-1, the outcome of which is a similar lack of microfibrils in the ECM [20,22].

The most common type of mutations are missense mutations that result in the disruption of a cysteine residue [9,24]. The fibrillin-1 protein has several repeated domains including 47 epidermal growth factor (EGF)-like domains, 43 of which are involved in calcium-binding (cbEFG-like), seven TGF-β binding protein-like (TB) domains and two hybrid domains [25,26]. Each of these domains are cysteine-rich with six to eight conserved cysteine residues that play a critical role in the folding and stability of the fibrillin-1 protein [25,27]. Mutations affecting a conserved cysteine have been shown to either increase the susceptibility of fibrillin-1 to proteolysis [28,29] or disrupt the folding and secretion of fibrillin-1 leading to intracellular retention [30]. The outcome of either scenario is a similar decreased microfibril stability and abundance.

Following the discovery that mutations in fibrillin-1 result in TGF-β dysregulation, a research area emerged focussing on the antagonism of TGF-β as a therapeutic strategy [31,32]. However, no breakthroughs have yet been made in the field and treatment of MFS patients remains heavily focused on symptom management. The current standard of care includes surgical correction of scoliosis, ectopia lentis, pectus deformities and aortic dilatation, as well as pain management and the use of β-Blockers [33,34,35] or more recently angiotensin II receptor type 1 blockers [1,31,36] to slow aortic growth. Here, we propose that personalised medicines using antisense oligonucleotides (AO) to alter *FBN1* exon selection during the splicing process, may be an appropriate therapeutic approach for some individuals with Marfan syndrome.

Antisense oligonucleotides (AO) are short sequences, generally between 15 and 30 bases in length, that are single-stranded analogues of nucleic acids. An AO is designed to be complementary to the region of interest binding to the target RNA or DNA through Watson-Crick base pairing. When bound to the target sequence and depending upon the chemistry, AOs can alter transcripts through two main mechanisms; recruiting RNase-H to cleave the target leading to degradation [37,38] or physically blocking the binding of regulatory factors or machinery of the transcription [39], translation [40] or splicing [41,42] processes. Several studies have outlined the potential of AOs in the treatment of genetic diseases. Several notable examples—eteplirsen [43,44], nusinersen [45,46] and more recently golodirsen [47], viltolarsen [48] and casimersen [49]—have now been approved by the United States Food and drug authority (FDA). All four drugs are a class of AO commonly referred to as splice switching. Splice switching AOs function by blocking the splicing machinery or regulatory features, altering the normal splicing process.

Splicing is an essential process for all multi-exon genes; removing the non-coding introns and re-joining the coding exons, before the transcript can be translated. The splicing process is, therefore, tightly regulated by several *cis*- and *trans*-acting elements. The majority of multi-exon genes, however, also undergo a process called alternative splicing [50]. Alternative splicing allows the production of multiple transcripts, and thus proteins, from a single gene, significantly increasing genetic complexity and diversity. To maintain the precise removal of introns, as well as supporting alternative splicing, the regulation of these processes is multi-layered and complex while maintaining a level of flexibility in the definition of an exon. Utilising the inherent flexibility of exon definition AOs can be targeted to motifs involved in exon recognition and processing, such as the acceptor and donor splice sites, as well as hotspots for splicing enhancers either in the intron or exon. Targeting enhancer sites can block the binding of positive splicing factors, thus decreasing the definition and recognition of an exon sufficiently to result in its exclusion [51,52]. Contrariwise, targeting exonic splicing silencer or intronic splicing silencer sequences can inhibit the binding of negative splicing factors, increasing exon recognition leading to inclusion [52,53].

The affinity, specificity, efficiency, stability and tolerance of an AO can be increased by modifying the chemical structure of the monomers and the backbone. Two widely used chemistries were utilised in this study. First of which has 2′-O-methyl (2′OMe) ribose ring modifications on a negatively charged phosphorothioate (PS) backbone. The resulting 2′OMe-PS compounds are robust RNase-H independent AOs that are nuclease resistant and relatively cost-effective to synthesise. The second chemistry is the phosphorodiamidate morpholino oligomer (PMO) that completely replaces the ribose sugar moiety with a morpholine ring and has phosphorodiamidate linkages [54]. The PMO chemistry is RNase-H independent, and has a neutral charge that precludes interaction with proteins, greatly reducing the possibility of off-target effects [55,56]. While the PMO chemistry is both more technically challenging and costly to synthesise than 2′OMe-PS AOs, PMOs are generally recognised as both safe and effective in a clinical setting, making it a promising chemistry for drug development [43,57].

As described previously, fibrillin-1 monomers are secreted from the cell and rapidly aggregated into multimer units to form the backbone of fibrillin-1 microfibrils [10]. Mutations in *FBN1* disrupt the formation of microfibrils, ultimately leading to a disease phenotype; either Marfan syndrome or another type-1 fibrillinopathy. Therefore, we propose that removal of a mutation-associated exon from all *FBN1* transcripts during the splicing process could result in the production of fibrillin-1 proteins that are able to form functional microfibrils, restoring ECM stability. To assess the viability of this hypothesis, we addressed three preliminary questions using *FBN1* exon 52 as a model. (1) Can an exon be specifically removed from *FBN1* pre-mRNA using antisense oligonucleotides, (2) Can sufficient exon skipping be achieved, and (3) can the internally truncated fibrillin-1 protein form microfibrils.

## 2. Results

### 2.1. The FBN1 Transcript and Antisense Oligonucleotide Design

The fibrillin-1 transcript (LRG_778t1; NM_000138.4) contains 11,695 bases separated into 66 exons, 65 of which encode the 350 kDa fibrillin-1 protein (Figure 1a). Exon 52 of *FBN1* encodes a total of 22 amino acids and makes up a portion of the sixth, of seven TB domains. Over 20 disease-causing mutations have been reported to affect exon 52, the majority of which result, or are predicted to result, in aberrant exon 52 splicing [9,24]. To excise exon 52, along with the flanking introns during the splicing process, five AOs were designed to target the acceptor, and donor splice sites as well as exonic splice enhancer (ESE) and intronic splice enhancer (ISE) sites across exon 52, predicted using the webtool SpliceAid [58] (Figure 1c). SpliceAid examines the exonic and intronic sequence of interest and determines associated silencer and enhancer sites. Each site is given a score of 1 to 10 that indicates the strength of the site, with the value closest to 10 being the strongest ESE sites.

### 2.2. Evaluation of AOs to Induce Exon 52 Skipping from FBN1 Transcripts

Initial AO screening was performed using AOs composed of 2′OMe-PS molecules. An unrelated control AO that does not anneal to any transcript was included in all transfections as a sham treatment to observe any chemistry or delivery related effects on cell health and transcript abundance. A complete list of AOs can be found in Table 1.

All 2′OMe-PS AOs were transfected into fibroblasts, derived from a healthy control subject, at three concentrations (200 nM, 100 nM and 50 nM) and incubated for 24 h before collection for RNA extraction and RT-PCR analysis to assess exon skipping efficiencies. The 24-h transfection incubation period was chosen after a time course of 24, 48 and 72 h revealed negligible differences in skipping efficiency over time (data not shown). However, treated cells, in particular those treated with the higher AO concentration, showed changes in morphology and began to die after 48 h (data not shown).

Analysis of PCR amplicons revealed the presence of two products in several samples; the expected full length (FL) product between exon 47 forward and exon 54 reverse primers, as well as a smaller product corresponding to the expected size after skipping of exon 52 (Δ52) (Figure 2a). The identity of the amplicons was confirmed by band purification and Sanger sequencing, confirming the precise removal of all 66 bases of exon 52 (Figure 2b,c).

The most efficient exon 52 skipping was induced by AO52.1^n^, with 41% of transcripts lacking exon 52 after transfection at 200 nM (Figure 2a). However, three other sequences, AO52.1^m^, AO52.3 and AO52.5 were also relatively efficient, inducing up to 40%, 37% and 22% skipping, respectively (Figure 2a). The remaining two sequences did not induce any measurable exon 52 skipping. The sequences AO52.1^n^ and AO52.1^m^ differ by a single base, with each being an exact complementary pair for the wild-type and a known Marfan syndrome patient cell line, respectively. This AO was designed in the hopes of understanding the mechanism behind the mutation that is known to cause mis-splicing of exon 52. The one base-pair mismatch did not greatly reduce the efficiency of AO52.1^m^ in healthy control cells.

Following initial AO screening, removal of exon 52 was deemed an appropriate option. In an attempt to further enhance exon exclusion, two AOs targeting *FBN1* exon 52 were combined into cocktails and evaluated. This method has previously been shown to boost skipping efficiency through synergy between the two AOs [59]. Six of the eight cocktails tested, induced between 23% and 42% exon 52 skipping, suggestive of an additive or baseline effect, similar to that achieved with a single AO. The combination of AO52.1^n^ with AO52.2 or AO52.3 was antagonistic resulting in no measurable exon skipping (Appendix A). No synergistic cocktails were identified, therefore AO52.1^n^ was chosen as the most promising candidate and the sequence was synthesised as a PMO for further analysis (Table 1).

### 2.3. PMO52 Induces Efficient Exon 52 Skipping and an Increase in Fibrillin-1 Microfibrils Determined by Immunofluorescent Staining

To both confirm the efficiency of the AO52.1^n^ sequence as a PMO, and to assess the effect of exon 52 skipping on fibrillin-1 microfibril formation, PMO52 was transfected into healthy control fibroblasts. An electroporation-based transfection method, nucleofection, was used to deliver PMO52 into control cells at two concentrations, 250 and 50 µM, as calculated in a 20 µL cuvette. Transfected cells were plated either directly into 24 well plates or onto coverslips and incubated for 72 h before cells were collected for RNA analysis and coverslips fixed for immunofluorescent staining.

Representative results of healthy control cells treated with PMO52 are presented in Figure 3a. These representative RT-PCR results reflect exon 52 removal from approximately 100% of transcripts, with no measurable FL product remaining. The lower AO concentration was observed to induce approximately 50% skipping, with no endogenous exon 52 skipping observed in either the control or untreated samples (Figure 3a). Analysis of RT-PCR amplicons across four replicates revealed relatively consistent dose-dependent exon 52 skipping. On average Δ52 transcripts constituted 91% of total *FBN1* transcripts from cells transfected at the higher concentration and 55% at the lower concentration (Figure 3b). The lowest skipping efficiency at the highest concentration was 74%; in the same experiment, the lower concentration maintained the average 55% skipping efficiency (Figure 3b).

Fibrillin-1 protein was detected through immunofluorescent staining using a fibrillin-1 specific primary antibody and a fluorescently tagged secondary. Staining of the untreated sample revealed the long-thin extracellular fibre-like formations expected of fibrillin-1 (Figure 3c iii). Notably, the morphology of fibres in the sample with more than 90% skipping is trending toward those seen in the untreated healthy control (Figure 3c i,iii). The abundance of fibrillin-1 staining, as well as the abundance of fibre-like formations is, however, noticeably reduced in the 250 µM-treated samples. In contrast to the fibre formation in both untreated and high concentration of PMO52, healthy control cells treated with 50 µM of PMO52, present with a complete loss of fibrillin-1 fibres and an overall reduction in fibrillin-1 staining (Figure 3c ii).

Following successful exon 52 skipping in healthy control fibroblasts, PMO52 was further assessed in fibroblasts derived from an individual with Marfan syndrome obtained from the NIGMS Human Genetic Cell Repository at the Coriell Institute for Medical Research. The patient fibroblasts (MFS^Δ52^) were reported to harbour a silent c.6354C > T, p.(Ile2118Ile) mutation in *FBN1* that was found to result in the in-frame skipping of exon 52 [60]. PMO52 was nucleofected into the MFS^Δ52^ and healthy control fibroblasts and collected for RNA and protein analysis after 4 days. The PMO52 sequence resulted in robust skipping in both cell lines and a strong dose response was observed. Treatment with 50 µM resulted in 64% and 41% exon 52 skipping in MFS^Δ52^ and healthy control fibroblasts, respectively (Figure 4a). Increasing the concentration to 250 µM resulted in 92% skipping in both cell lines (Figure 4a).

Immunofluorescence once again revealed the strong fibre-like structures formed by fibrillin-1 in the untreated, and GTC treated, healthy control fibroblasts. These fibres were completely lost when 41% skipping was induced (Figure 4b vi). The staining pattern mirrored that observed in the untreated MFS^Δ52^ fibroblasts with no fibre formation and minimal diffuse fibrillin-1 staining (Figure 4b iv). The minor increase in the proportion of skipped products after treatment of MFS^Δ52^ fibroblasts with 50 µM of PMO52, did not alter the fibrillin-1 staining pattern (Figure 4b ii). In both cell lines, treatment with 250 µM of PMO52 resulted in the formation of fibrillin-1 fibres. In the healthy control cells, these fibres, while of high staining intensity, were fragmented and reduced in abundance compared to the untreated healthy control sample (Figure 4b v,viii). However, in MFS^Δ52^ cells treated with the 250 µM, fibrillin-1 fibres had a continuous, non-fragmented morphology trending toward those seen in the untreated healthy control (Figure 4b i,viii). The fibres are also relatively abundant filling the majority of the field-of-view; however, they are not as plentiful as those seen in the untreated healthy control that form a multi-layered lattice (Figure 4b i,viii). Similar staining patterns were seen across multiple biological replicates representative images of one replicate are presented in Appendix A.

## 3. Discussion

Although Marfan syndrome is well established as an inherited connective tissue disorder caused by mutations in the fibrillin-1 gene, the exact mechanism of pathogenesis has not been fully resolved. Current understanding suggests that the pathogenesis is dependent on the mutation type with an overarching basis that a lack of functional fibrillin-1 microfibrils leads to *TGF-β* dysregulation, further compounding ECM destabilisation [15,61]. Therefore, we propose that removal of a mutation-associated exon from all *FBN1* transcripts could result in the production of internally truncated fibrillin-1 proteins that retain some function and are able to form microfibrils. We addressed this hypothesis by designing antisense oligonucleotides to induce exon 52 exclusion from unaffected *FBN1* pre-mRNA. We suggest that many fibrillin-1 gene lesions will be amenable to the removal of the affected exon for two main reasons. Firstly, fibrillin-1 is highly repetitive suggesting the possibility of functional redundancy. Secondly, excluding exons 2, 3, 64, 65 and 66, the majority of fibrillin-1 exons are in-frame, and therefore can be removed without disrupting the reading frame.

Here, we describe evidence for the efficient removal of *FBN1* exon 52. Of the five 2′OMe-PS AOs tested three were found to induce exon 52 skipping. Earlier dystrophin screening studies similarly found that two out of three AOs induced some skipping, albeit at different efficiencies [59]. The different delivery methods, concentration and length of incubation that were used when assessing the 2′OMe-PS and PMO sequences, make it impossible to directly compare their efficiencies. However, our data show that the proportion of Δ52 transcripts induced is greater when using the PMO sequence, this is likely in-part because a higher PMO concentration and longer incubation period could be used without a large decrease in cell viability. A higher concentration is generally required for the PMOs as the neutral charge of the chemistry hinders cellular uptake [62,63]. Immunofluorescent staining of fibrillin-1 in treated cells supports the hypothesis that fibrillin-1^Δ52^ proteins can interact to form multimers. We observed fibre formation, mirroring that of the untreated control cells, when more than 92% exon 52 skipping was induced in either the healthy control or MFS^Δ52^ patient fibroblasts. We also established that inducing approximately 50% exon 52 skipping results in a complete loss of fibrillin-1 fibre staining in healthy control fibroblasts mimicking the disease-like state caused by splicing mutations. Together, these results demonstrate proof-of-concept that the internally truncated fibrillin-1^Δ52^ proteins produced through efficient exon 52 skipping are able to form multimers.

Of particular interest, is the total loss of microfibril formation that results from the induction of a combination of wild-type and *FBN1*^Δ52^ transcripts after sub-optimal levels of PMO-induced exon skipping. This finding demonstrates the inability of the heterogenous population of Δ52 fibrillin-1 proteins to form microfibrils, supporting the dominant-negative pathogenic mechanism [64]. We observed a similar lack of extracellular fibrillin-1 as that reported by Liu et al. [65] resulting from the c.6354C > T mutation that leads to 41% fibrillin-1 synthesis and only 5% deposition of fibrillin-1 in the extracellular matrix. However, this finding also has relevance to mapping of amenable *FBN1* exons that could be targeted in a splice intervention therapy. The elimination of microfibrils and subsequent formation with increased skipping efficiency could prove to be an invaluable tool in optimisation of fibrillin-1 AO sequences as well as identification of potential target exons for therapeutic intervention. Importantly, being able to induce a disease-like state would allow the use of healthy control cells, rather than specific patient cells, for the identification of exons that when removed do not affect the expression or function of fibrillin-1. Furthermore the occurrence and severity of dominant negative effects depend on the mutation type and location [66]. For example, duplications causing a mouse model of tight skin syndrome, result in a larger fibrillin-1 protein that has been shown to only form microfibrils in the presence of wild-type fibrillin-1 [67]. While the co-polymerisation of the two fibrillin-1 isoforms forms functionally deficient microfibrils, the outcome is a mild phenotype lacking vascular involvement [67]. Identification of such naturally occurring co-polymerisation events that lead to a mild phenotype could reveal potential therapeutic strategies to assess in the future.

We predict that the AOs reported here can manipulate *FBN1* splicing such that, at lower skipping efficiencies disease characteristics can be induced in unaffected cells and upon increased efficiency, sufficient skipping can be induced to reduce the key phenotype of MFS. This prediction is based on the dominant-negative model that suggests that in the presence of two protein isoforms, the aberrant protein disrupts the formation of microfibrils by the wild-type protein [18]. It is unknown exactly what ratio of wild-type to aberrant proteins would negate the dominant negative effects. However, the results presented here suggest that this tolerable threshold may be approximately 95%. We demonstrate that fibrillin-1 fibres can be formed when exon 52 skipping is sufficient, likely >90% skipping, such that more than 95% of total multimers that are formed would be of the fibrillin-1^Δ52^-fibrillin-1^Δ52^ structure. We also note, however, that *FBN1* mutations resulting in haploinsufficiency lead to disease. Therefore, we believe that the abundance of microfibrils has to be maintained, as a minimum, above that observed in haploinsufficiency patients [19,21]. It is also important to note that while this exon skipping strategy relies on excluding the target exon from both the mutation-harbouring and healthy *FBN1* transcripts, the exon skipping is at the mRNA level, and therefore not permanent as would be the case for other techniques such as gene therapy.

While we demonstrated efficient and consistent exon skipping using PMO52, the concentrations used are relatively high, when compared to similar studies targeting other genes [59,68]. One of the possible explanations for the high concentration required is the abundance of fibrillin-1 transcripts. Fibrillin-1 RNA is expressed in the vast majority of cell types and is maintained at relatively high levels throughout the body [69]. We noted the efficient PCR amplification of *FBN1* transcripts; requiring 20 or fewer rounds of amplification coupled with the need for very low template concentrations (25 ng). While the in vitro PMO transfection concentrations used seem relatively high, we noticed no changes in morphology or health of cell cultures up to four days post-transfection with PMO52. The PMO chemistry is generally considered to be safe with no off-target effects nor toxicity [43,57,62]. Nevertheless, while the PMO chemistry may be safe and a higher concentration required due to the level of fibrillin-1 expression, there are still several ways in which the efficiency of an AO can be improved, including optimisation of AO delivery, sequence, length and chemistry. Further optimisation could allow for the use of a significantly lower dosage that in turn would not only reduce the possibility of off-target effects or toxicity but also lower the cost of treatment.

The fibrillin-1 protein produced by excising exon 52 is predicted to be internally truncated, fibrillin-1^Δ52^, and lack the last seven amino acids of the sixth TB domain. This isoform has only been reported in the context of exon 52 mutations, where it is known to act in a dominant-negative manner against the wild-type protein and result in a severe lack of functional microfibrils [60,65]. Liu et al. [65] also reported that an exon 52 splicing mutation leads to reduced fibrillin-1 synthesis, less than 50% of that observed in healthy controls, while the mutant mRNA levels remain unchanged, suggesting that the fibrillin-1^Δ52^ proteins are unstable. Liu et al. [65] suggests this instability may result from the partial deletion of a TB domain that leads to misfolding of the fibrillin-1 protein increasing its susceptibility to proteolysis. If this is the case then attempts to induce exon skipping of other exons encoding partial TB domains; 10, 11, 17, 18, 38, 39, 42, 43 and 51, would likely face the same issue. It is possible that removal of the two exons encoding the TB domain as a pair could solve this problem. However, our findings suggest that the fibrillin1^Δ52^ proteins produced through exon 52 skipping are able to be synthesised, secreted from the cell and form fibre-like structures in both healthy control and MFS^Δ52^ patient fibroblasts. Nevertheless, the synthesis, deposition and function of fibrillin-1^Δ52^, especially in the absence of wild-type fibrillin-1, needs to be investigated further.

Here, we illustrate that when fibrillin-1^Δ52^ is the predominant product it is able to be both synthesised and secreted from the cell, with no evidence of intracellular staining. We also demonstrate that fibrillin-1^Δ52^ proteins can form fibres, provided that *FBN1^Δ52^* transcripts make up more than 90% of total *FBN1* transcripts. These results suggest that the fibrillin-1^Δ52^ protein is at least partially functional, although further protein analysis is required to assess if the fibres formed can interact with the microfibril associated proteins with which fibrillin-1 naturally interacts. The ability of fibrillin-1^Δ52^ proteins to sequester TGF-β also needs to be assessed. If removal of exon 52 disrupts the regulation of TGF-β, then regardless of the high skipping efficiency and fibre formation that is observed, symptoms such as aortic growth would continue to progress with no benefit from this treatment. To assess the effect of *FBN1* exon skipping and the function of the induced fibrillin-1 isoform especially the impacts on TGF-β signalling, surrogate markers such as the phosphorylation of Smad2 can be analysed [70,71]. The level of active and total TGF-β can similarly be assessed as a measure of functionality [61,71,72]. Such assays were outside the scope of the current study and will be the focus of further research.

As previously noted, our results support the hypothesis that fibrillin-1^Δ52^ can form fibres. However, while the morphology of fibres is superficially similar to that of the untreated, healthy control, their abundance is reduced. The reduction in abundance could be the result of the experimental design and protocols. For example, nucleofection can cause cell stress potentially reducing fibrillin-1 expression, or the transfection incubation time could be insufficient to allow more efficient formation of microfibrils after treatment. However, it is likely that, as reported by Liu et al. [65], fibrillin-1^Δ52^ synthesis is reduced in comparison to the wild-type. Western blotting analysis of intracellular and extracellular fibrillin-1 was attempted, however, due to poor signal and resolution in samples from healthy control fibroblasts we were unable to confirm any changes in fibrillin-1 abundance. Further optimisation of the Western blot protocol to produce reliable results is required before the effect of *FBN1* exon skipping on the abundance of fibrillin-1 can be assessed. While restoring microfibril abundance and function to a ‘normal’ state would be ideal, this may not be possible. We believe that any increase in functional microfibrils could provide a therapeutic benefit by reducing disease progression and severity.

As discussed earlier, the current standard of care for individuals living with MFS relies heavily on invasive surgical interventions and the lifelong use of medicines such as β-adrenergic receptor blockades that slow the progression of aortic dilation [33,34,35]. These interventions have proven lifesaving, as well as life-extending [5,73]. However, the burden of MFS on quality of life, and the economic stress, for both patients and their families, remains substantial [74,75]. In more recent years major efforts have been made to discover and develop therapeutics for MFS [31,32,33,76]. Research has focused on slowing aortic growth as well as a continued improvement upon current treatment strategies for the main symptoms of MFS. With FDA approval of AO therapeutics to restore gene function in spinal muscular atrophy [45] and Duchenne muscular dystrophy [77,78,79], we suggest that antisense oligonucleotide-mediated splice switching as described here could be an appropriate direction for the development of therapies for Marfan syndrome.

In conclusion, this study assessed the ability of a suite of AOs to induce targeted exon 52 skipping from full-length *FBN1* mRNA transcripts expressed in healthy control fibroblasts. The most efficient sequence, and the consequences of splice modification, was further evaluated in both healthy control and MFS^Δ52^ patient fibroblasts. We showed in vitro, that AO52.1^n^, AO52.1^m^ and AO52.3 as well as PMO52 induced dose-dependent exon 52 skipping. Encouragingly the presence of more than 90% of one transcript type; wild-type or *FBN1*^Δ52^, corresponded with the formation of fibrillin-1 fibres in both cell lines. In contrast, a mixed transcript pool resulted with the complete loss of fibrillin-1 fibres, mimicking the disease-like state.

While this study is a preliminary, in vitro investigation, our candidate PMO consistently induces efficient exon 52 exclusion while maintaining fibrillin-1^Δ52^ fibre formation. With increasing numbers of AO therapeutics being approved for clinical use, our results suggest that PMO52 may be an attractive therapeutic option for the treatment of Marfan syndrome caused by mutations in fibrillin-1 exon 52. This study provides proof-of-concept and a foundation for the further development of antisense oligonucleotide therapies for Marfan syndrome.

## 4. Materials and Methods

### 4.1. Design and Synthesis of Antisense Oligonucleotides

Antisense oligonucleotides were designed to target splicing regulatory motifs at the exon-intron junctions as well as exonic splicing enhancer sequences predicted using the SpliceAid web tool [58]. AO sequences were also analysed using NCBI nucleotide BLAST (NCBI, Bethesda, MD, USA) [80] to identify any possible off-target annealing. Antisense oligonucleotides with 2′OMe-PS chemistry were purchased from TriLink biotechnologies (Maravai LifeSciences, San Diego, CA, USA), and PMOs were purchased from GeneTools LLC (Philomath, OR, USA). The nomenclature of AOs is based on that described by Mann et al. [42] and provides information on the gene, exon, annealing co-ordinates and species. A full list of AOs used in this study are provided in Table 1. The *FBN1* exon nomenclature was determined with respect to the NCBI Reference Sequence NM_000138.4, in which the translation start codon is in the second of 66 exons.

### 4.2. Cell Culture and Transfection 

Fibroblasts were originally sourced from a dermal biopsy derived from a healthy volunteer with informed consent. The following cell line was obtained from the NIGMS Human Genetic Cell Repository at the Coriell Institute for Medical Research: GM21941 (Camden, NJ, USA). The use of human cells for this research was approved by the Murdoch University Human Ethics Committee, approval numbers 2013_156 (25 October 2013) and 2017_101 (12 May 2017) and The University of Western Australia Human Research Ethics Committee, approval number RA/4/1/2295 (21 April 2009). Healthy control fibroblasts were maintained in Dulbecco’s Modified Essential Medium (DMEM, Gibco; Thermo Fisher Scientific, Melbourne, Australia) supplemented with 10% foetal bovine serum (FBS, Scientifix, Melbourne, Australia). The MFS^Δ52^ patient fibroblasts were maintained in DMEM supplemented with 15% FBS (Scientifix, Melbourne, Australia) and 1% glutaMax (Gibco; Thermo Fisher Scientific, Melbourne, Australia). Both cell lines were maintained at 37 °C with 5% CO_2_.

Antisense oligonucleotides (2′OMe-PS chemistry) used for target site screening were transfected into healthy control fibroblasts using Lipofectamine 3000 (L3K, Life Technologies, Melbourne, Australia). Transfections were prepared by incubating the AO with 3 µL of L3K, at room temperature in 50 µL of Opti-MEM (Gibco; Thermo Fisher Scientific, Melbourne, Australia), according to manufacturer’s protocol. The transfection mixture was then diluted to the desired AO concentration in a final volume of 1 mL and applied to cells. Transfected cells were incubated for 24 h before collection.

The PMO was delivered using the 4D-Nucleofector™ and P3 nucleofection kits (Lonza, Melbourne, Australia). One microliter of stock PMO (5 mM), either undiluted (250 µM) or diluted 1:4 in sterilised water (50 µM), was added into a cuvette along with 300,000 fibroblasts resuspended in 19 µL of pre-warmed transfection solution. The mixture of fibroblasts and PMO was subsequently nucleofected using the pulse code, CA 137, previously optimised in our laboratory for dermal fibroblasts. Nucleofected fibroblasts were maintained in DMEM supplemented with 5% FBS before collection after 3 or 4 days.

### 4.3. RNA Extraction and RT-PCR Analysis

Total RNA was extracted using MagMax™ nucleic acid isolation kits (Thermo Fisher Scientific, Melbourne, Australia) as per the manufacturer’s protocol. Total RNA concentration and purity were determined using a Nanodrop 1000 spectrophotometer (Thermo Fisher Scientific, Melbourne, Australia). Transcripts were amplified using one-step SuperScript^®^ III reverse transcriptase (Thermo Fisher Scientific, Melbourne, Australia) with 25 ng of total RNA as a template. To assess exon 52 skipping, *FBN1* transcripts were amplified using exon 47 Forward (5′-GGTTTCATCCT-TTCTCACAAC-3′) and exon 54 Reverse (5′-TCACATGTCATCATTGGACC-3′) primers (Integrated DNA Technologies, Sydney, Australia). The cycling conditions were as follows; 55 °C for 30 min, 94 °C for 2 min followed by 20 cycles of 94 °C for 30 s, 55 °C for 30 s and 68 °C for 1 min. The PCR amplicons were fractionated on 2% agarose gels in Tris-Acetate-EDTA buffer. Relative exon skipping efficiency was estimated through densitometric analysis of images using ImageJ (version 1.8.0_112) imaging software (NIH, Bethesda, MD, USA) [81] and reported as the proportion of FL or Δ52 transcript products relative to the sum of products.

### 4.4. Immunofluorescence 

Immediately after nucleofection, 100,000 fibroblasts were seeded into each well of a 24-well plate lined with a 13 mm #0 round uncoated glass coverslip. Cells were incubated for 72 h before being fixed in ice-cold acetone: methanol (1:1, *v*:*v*) and allowed to air dry. Fixed cells were washed once with PBS to rehydrate before blocking with 10% goat serum in PBS for 1 h at room temperature. The primary antibody, Anti-fibrillin-1 antibody clone 26 (Merck Millipore, Sydney, Australia), was applied at a dilution of 1:100 in 1% goat serum-PBS and incubated overnight at 4 °C. Secondary antibody; AlexaFluor568 anti-mouse IgG (Thermo Fisher Scientific, VIC, Australia) was applied, 1:400, for 1 h at room temperature, and co-stained with Hoechst 33,342 (Sigma-Aldrich, Sydney, Australia) for nuclei detection (1 mg/mL diluted, 1:125). Coverslips were mounted using ProLong™ Gold antifade mountant (Thermo Fisher Scientific, Melbourne, Australia). Fibrillin-1 was detected using a Nikon 80i microscope with NIS-Elements software (Nikon, Adelaide, Australia). The brightness and contrast of individual channel images were altered equally for each image, then merged. The merged image was cropped from the original 1280 × 1024 pixel image using Adobe Photoshop CC. A 20 µm scale bar was added using ImageJ software (NIH, Bethesda, MD, USA) [81].

## Figures and Tables

**Figure 1 ijms-22-03479-f001:**
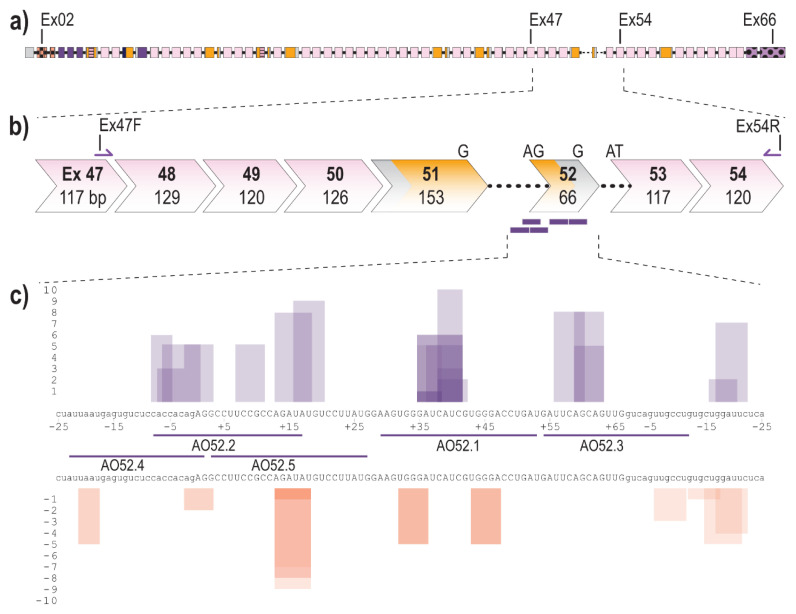
Schematic of *FBN1* pre-mRNA highlighting the region between exons 47 and 54 as well as AO binding sites (**a**) Full fibrillin-1 pre-mRNA transcript with each box representing an exon. The solid black line represents introns (not to scale); (**b**) highlighting the region between exons 47 and 54; showing forward and reverse primers and antisense oligonucleotide binding sites (purple bars). Chevron sides indicate exons bounded by partial codons. Pink and yellow fill indicate regions encoding cbEGF-like and TB domains, respectively. The black dotted line indicates partial introns 51 and 52.; (**c**) Antisense oligonucleotide binding sites and the regulatory motifs they target predicted using spliceAid [58]. Each box represents a predicted enhancer (purple) or silencer (orange) site. The height of the box represents the strength of the site with 1 being the weakest and 10 being the strongest. Exonic and intronic sequences are shown in upper- and lower-case, respectively.

**Figure 2 ijms-22-03479-f002:**
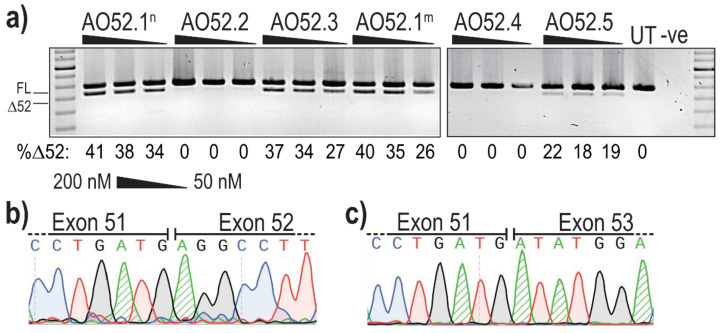
Evaluation of AOs designed to induce *FBN1* exon 52 skipping. (**a**) Screening of 2′OMe-PS AOs targeting exon 52. Healthy control fibroblasts were transfected with AOs as lipoplexes at three concentrations, 200, 100 and 50 nM. The values below each gel image indicate the percentage of exon 52 skipped (Δ52) transcripts in each sample. Ctrl: an unrelated sequence used as a sham treatment, UT: untreated control, -ve: RT-PCR negative control. 100 bp molecular marker used for size reference. The gels were cropped for presentation. Full gel images are presented in Appendix A; (**b**) Sanger sequencing analysis showing the junction between exons 51 and 52 in full-length transcripts (FL, 859 bp); (**c**) Sanger sequencing analysis showing the junction between exon 51 and 53 exon 52-skipped transcripts (Δ52, 793 bp).

**Figure 3 ijms-22-03479-f003:**
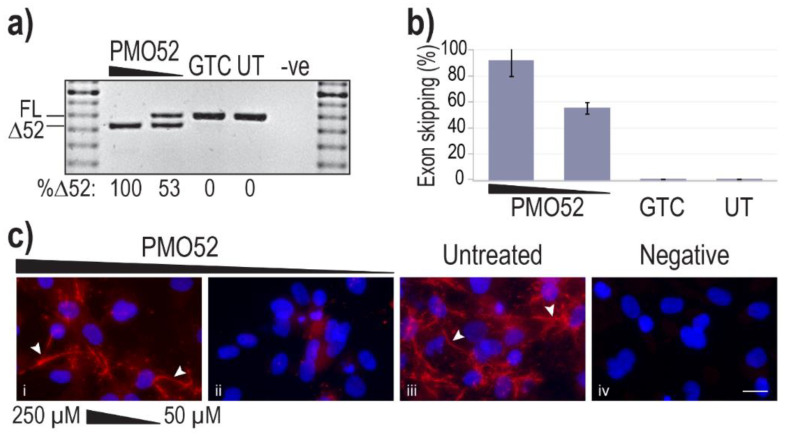
Efficiency and effect of PMO52. Healthy control fibroblasts were harvested for protein and RNA analysis, 72 h after nucleofection with PMO52 at concentrations of 250 µM and 50 µM; calculated in a 20 µL cuvette. (**a**) Agarose gel fractionation of *FBN1* exons 47 to 54 amplicons showing full-length (FL, 859 bp) and exon 52-skipped (Δ52, 793 bp) transcripts. The values below the gel image indicate the percentage of Δ52 transcripts. GTC: Gene Tools control PMO, UT: Untreated control, -ve: RT-PCR negative control, 100 bp molecular marker used as a size reference. The gels were cropped for presentation. Full gel images are presented in Appendix A (**b**) The percentage of full-length transcript relative to total *FBN1* transcript across four biological replicates (means plus error bars. Error bars = standard deviation, *n* = 4). (**c**) Fibrillin-1 protein analysed via immunofluorescent staining. Merged fluorescence images of Hoechst staining of the nucleus (blue) and fibrillin-1 (red) with ‘healthy’ fibre-like morphology of fibrillin-1 indicated by white arrowheads. Negative: no primary antibody added, to control for non-specific binding of the secondary antibody. Untreated: no PMO added. Scale bar = 20 µm. The images were cropped for presentation. Full images are presented in Appendix A.

**Figure 4 ijms-22-03479-f004:**
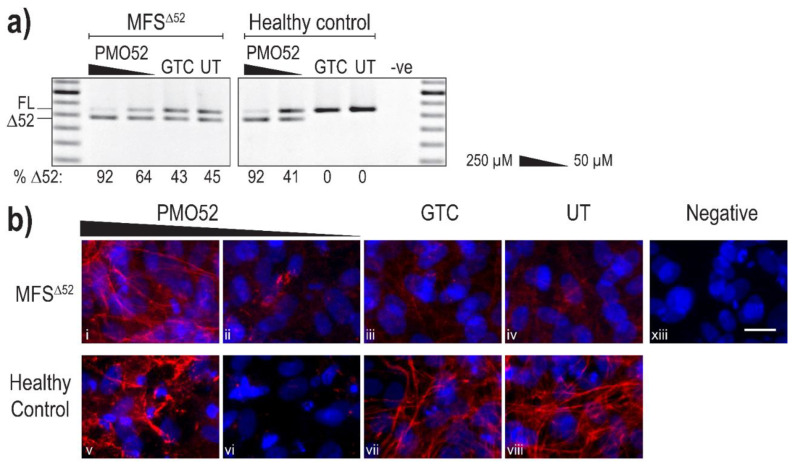
Evaluation of PMO52 in MFS^Δ52^ fibroblasts. Healthy control and MFS^Δ52^ patient fibroblasts were transfected with PMO52 (250 μM and 50 μM), GTC or left untreated. Cells were collected 4 days post-transfection. (**a**) RT-PCR analysis of *FBN1* exons 47 to 54 amplicons showing full-length (FL, 859 bp) and exon 52-skipped (Δ52, 793 bp) transcripts. The relative abundance (%) of Δ52 amplicons are shown below each gel image. GTC: Gene Tools control PMO, UT: untreated control, -ve: RT-PCR negative control, 100 bp molecular marker used as a size reference. The gels were cropped for presentation. Full gel images are presented in Appendix A (**b**) Representative images showing fibrillin-1 (red) and nuclei (blue) in MFS^Δ52^ or healthy control fibroblasts treated with (**i,v**) 250 µM of PMO52, (**ii,vi**) 50 µM of PMO52, (**iii,vii**) 250 µM of GTC or (**iv,viii**) left untreated. (**xiii**) negative control with no primary antibody added. Scale bar = 20 µm. The images were cropped for presentation. Full images are presented in Appendix A.

**Table 1 ijms-22-03479-t001:** Antisense oligonucleotide binding coordinates and sequences.

Nomenclature (FBN1 H…)	Name	Sequence (5′-3′)	Chemistry
52A(+29+53)N	AO52.1^n^	AUC AGG UCC CAC GAU GAU CCC ACU U ATC AGG TCC CAC GAT GAT CCC ACT T	2′OMe-PS PMO
52A(+29+53)M	AO52.1^m^	AUC AGG UCC CAC AAU GAU CCC ACU U	2′OMe-PS
52A(−08+17)	AO52.2	UAU CUG GCG GAA GGC CUC UGU GGU G	2′OMe-PS
52D(+13-12)	AO52.3	CAG GCA ACU GAC CAA CUG CUG AAU C	2′OMe-PS
52A(−23+02)	AO52.4	CUC UGU GGU GGA GAC ACU CAU UAA U	2′OMe-PS
52A(+03+27)	AO52.5	CAU AAG GAC AUA UCU GGC GGA AGG C	2′OMe-PS
Control AO	Ctrl	GGA UGU CCU GAG UCU AGA CCC UCC G	2′OMe-PS
GeneTools Control	GTC	CCT CTT ACC TCA GTT ACA ATT TAT A	PMO

## Data Availability

Not applicable.

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
