# Peer review of "Proof-of-Concept: Antisense Oligonucleotide Mediated Skipping of Fibrillin-1 Exon 52"

_ijms, 2021, doi:10.3390/ijms22073479_

Round 1

Reviewer 1 Report

This paper used antisense oligonucleotides (AOs) to induce skipping of exon 52 (52Δ) in fibrillin-1 (FBN1) gene transcription. The authors compared different predicted sites and combinations with 2′OMe ribose modified AOs. The most efficient site (AO52.1n) was chosen. Furthermore, phosphorodi-amidate morpholino oligomer (PMO) modified AO52.1n, which has higher binding efficacy than 2′OMe modified, was used for transfection. PMO-AO52.1n presented 53 % 52Δ with 50 uM, and 100% removal at a transfection concentration of 250 uM. The authors stated that fibrillin-1 fiber formation was inhibited by 53% 52Δ, and rescued by 100% 52Δ in healthy fibroblasts. Similar results were obtained using an 52Δ mutant MFS fibroblasts. The authors proposed that personalised medicines using AO to alter FBN1 exon selection during the splicing process, may be an appropriate therapeutic approach for some individuals with Marfan syndrome. While the data are interesting, several aspects of the paper were not justified and supported by the results.

Comments (major):

  1. The fibrillin-1 immunofluorescence staining of the healthy control in fig. 3 and 4 does not support the statement, that “the morphology of fibres in the sample with more than 90% skipping is comparable to those seen in the untreated healthy control”, given the low signal/noise ratio of the staining, the few fibers relative to the control, and the bias picking zoomed-in regions. Quantification of the fibrillin microfibrils (intensity and fiber length) among different repeats using multiple images are missing. Also, biological replicates of normal controls are not included and analyzed. Quantifications are also not provided for the MFS fibroblasts.
  2. The author concluded that less fibers was the result of inability of multimerization between full length fibrillin-1 and the exon-skipped fibrillin-1. However, the authors did not determine the fibrillin-1 mRNA and protein expression levels, comparing the treated to the control groups, using qPCR, and western blot of conditioned medium. Therefore, the conclusion is not justified. Additionally, how do the authors reconcile the data with the fact that the much larger Tsk fibrillin 1 can copolymerize with wild-type fibrillin 1 (PMID 10931876). This is not the same region in fibrillin-1 than exon 52, but it shows the principle that a larger fibrillin-1 with more exons can co-polymerize with a shorter one.
    In the discussion, the authors pointed out the inability of heterogenous fibrillin-1 to form fibers. However, there is still around 50% of normal fibrillin-1 present in the MFS fibroblasts. The authors mention a possible dominant negative effect, but did not further allude how this effect could work in regard to exon 52.
  3. The efficiencies of the transfection of the AOs is not shown or controlled. Thus, it is not valid to conclude that one AO is more efficient than others on exon skipping. It may at least partially result from different transfection efficiencies.
  4. The authors analyzed one mutation skipping exon 52, and proposed conceptually personalized medicine for Marfan syndrome (MFS). How many patients do have this particular mutation, probably only 1-2 total worldwide? Given the low number of MFS patients harboring this mutation or possibly a few others in exon 52, it is very likely not possible to perform clinical trials to test this approach. In this sense the provided argumentation is not realistic.

Author Response

Reviewer 1

Comments and Suggestions for Authors

Comments (major):

  • a) The fibrillin-1 immunofluorescence staining of the healthy control in fig. 3 and 4 does not support the statement, that “the morphology of fibres in the sample with more than 90% skipping is comparable to those seen in the untreated healthy control”, given the low signal/noise ratio of the staining, the few fibers relative to the control, and the bias picking zoomed-in regions.

We have changed this sentence to read “the morphology of fibres in the sample with more than 90% skipping is “trending towards” those seen in the untreated healthy control” to indicate that the change is not completely normal but better than 50% skipping and the near complete lack of staining in patient cells.

Line:  236

The similar sentence referring to Fig 4 has been changed to read “fibrillin-1 fibres had a continuous, non-fragmented morphology trending toward those seen in the untreated healthy control”

Line: 278-279

To address the bias picking comment, the entire image was included in the supplementary material (Figure S3).

  1. b) Quantification of the fibrillin microfibrils (intensity and fiber length) among different repeats using multiple images are missing.

We are uncertain how to measure fibre length as this is a lattice/mesh. Measuring intensity based on immunostaining is very subjective and does not appear to be commonly presented for fibrillin-1.

  1. c) Also, biological replicates of normal controls are not included and analyzed. Quantifications are also not provided for the MFS fibroblasts.

Replicate experiments (n=3) were undertaken and representative images were shown in the manuscript

  • a) The author concluded that less fibers was the result of inability of multimerization between full length fibrillin-1 and the exon-skipped fibrillin-1.

We are stating one of the pathogenic mechanisms is dominant negative, as cited by many other papers, which we acknowledge is clearly dependent on mutation type and location.

  1. b) However, the authors did not determine the fibrillin-1 mRNA and protein expression levels, comparing the treated to the control groups, using qPCR, and western blot of conditioned medium. Therefore, the conclusion is not justified.

The RT-PCR was carried out with 20 cycles of amplification, well within semi-quantitative range. Since amplicon signals from normal, patient, treated and untreated cells were similar, we may assume that the mRNA levels have not changed substantially. Since the skipping of exon 52 does not disrupt the reading frame, nonsense mediated decay of the mRNA should not be a factor.

We did attempt western blotting from conditioned media and cells (healthy and patient) but were unable to generate reliable and consistent signals.

  1. c) Additionally, how do the authors reconcile the data with the fact that the much larger Tsk fibrillin 1 can copolymerize with wild-type fibrillin 1 (PMID 10931876). This is not the same region in fibrillin-1 than exon 52, but it shows the principle that a larger fibrillin-1 with more exons can co-polymerize with a shorter one.

As we mentioned previously, the nature and location of the mutation will influence the pathogenic mechanism. The loss of exon 52 has been previously reported to be dominant negative with reduced extracellular deposition of fibrillin-1 (PMID 11710961 and 9241263). We did not discuss other FBN1 mutations, including those in animal models.

  1. d) In the discussion, the authors pointed out the inability of heterogenous fibrillin-1 to form fibers. However, there is still around 50% of normal fibrillin-1 present in the MFS fibroblasts. The authors mention a possible dominant negative effect, but did not further allude how this effect could work in regard to exon 52.

Liu et al (2001) BMN Med Genet 2:11 (PMID 11710961) reported that this patient had 41% synthesis of fibrillin-1 with normal mRNA levels but only 5% deposition, consistent with a Type II (PMID: 8040255) dominant negative mechanism. We revised discussion section to include this information

Line: 325-331

  • The efficiencies of the transfection of the AOs is not shown or controlled. Thus, it is not valid to conclude that one AO is more efficient than others on exon skipping. It may at least partially result from different transfection efficiencies.

The same transfection conditions were employed appropriate for the chemistry: Lipofectamine 3000 for the 2′-O methyl modified bases on a phosphorothioate backbone and nucleofection for the uncharged phosphorothioate morpholino oligomers.  There were multiple biological and technical replicates undertaken and we are confident that any variation in exon skipping efficiencies (as assessed in titration experiments at 200, 100 and 50 nM for the 2′OMe-PS AOs) is due to AO design and annealing.

  • The authors analyzed one mutation skipping exon 52, and proposed conceptually personalized medicine for Marfan syndrome (MFS). How many patients do have this particular mutation, probably only 1-2 total worldwide? Given the low number of MFS patients harboring this mutation or possibly a few others in exon 52, it is very likely not possible to perform clinical trials to test this approach. In this sense the provided argumentation is not realistic.

There have been 8 reports of this particular synonymous mutation from 4 independent laboratories.  In addition, there are different acceptor or donor splice site mutations that should lead to FBN1 exon 52 skipping, and furthermore other mutation types (nonsense/missense, frameshifting indels) could benefit from exon skipping if sufficient levels of exon removal can be achieved.  As mentioned in the title, this study was a proof-of-concept study and the data  is still preliminary requiring further development, as mentioned in the concluding paragraph.

Reviewer 2 Report

 Proof-of-concept: Antisense oligonucleotide mediated skipping of Fibrillin-1 exon 52

This is a highly interesting manuscript with great novelty, which provides new insights into the complex set of FBN1 mutations that all cause Marfan Syndrome. I still have a few suggestions.

It is true that there is very limited therapeutic options in Marfan Syndrome, however lately the use of angiotensin II receptor type 1 blockers (ARBs) is quite popular with Marfan patients and their physicians after a few reports on reduced aneurysm growth and aortic events. So the use of only beta-blockers is a bit outdated. This should be added in the introduction. Yet, if this is also symptoms management or actually interfering with the pathological process remains to be seen. (Circ Cardiovasc Genet. 2015 Apr;8(2):383-8. doi: 10.1161/CIRCGENETICS.114.000950. ; Lancet. 2019 Dec 21;394(10216):2263-2270. doi: 10.1016/S0140-6736(19)32518-8. ; Eur Heart J. 2020 Nov 14;41(43):4181-4187. doi: 10.1093/eurheartj/ehaa377. )

When searching the literature on Marfan fibroblasts it seems that many predicted dominant negative (DN) FBN1 mutations lead to impaired fibrillin-1 fiber formation, thus a haploinsufficient (HI) phenotype. It is important to actually measure fiber deposition because one cannot rely on predictions. This is shown here again and could be stressed more. The authors have shed light on a possible principle for this phenomenon, that the ratio between two different fibrillin-1 molecules 1:1 gives HI, while a ratio of 2:0 or 0:2 will rescue the imbalance.

Yet, many DN mutations are point mutations and thus to correct this, both the exon containing the mutation plus the wild type exon need to be deleted to rescue the phenotype probably, because two “misfolded” proteins may not rescue the fibers. I wonder if this would ever be ethical, to also change the normal FBN1 allele. Alternatively, enhancing fibrillin-1 expression to generate more wild type fibrillin-1 may overcome the imbalance as demonstrated by Judge et al (J Clin Invest . 2004 Jul;114(2):172-81. doi: 10.1172/JCI20641.) where they add an additional wild type fbn1, which could rescue the Fbn1 C1039G/+ marfan mouse phenotype. In a way this is similar since here they change the ratio from 1:1 to 2:1. Even an alternative Fbn1 mutant (C1663R) could rescue the C1039G/+ phenotype partially. Moreover, HI Marfan patients with low expression of wild type FBN1 had a more severe aortic phenotype compared to patients with high FBN1 expression (Hum Mol Genet. 2015 May 15;24(10):2764-70. doi: 10.1093/hmg/ddv037.). While I think their technique indeed shows proof of concept, which is a very novel way of thinking and highly interesting, I think the applicability should be downsized a bit in the manuscript.

Author Response

Reviewer 2

Comments and Suggestions for Authors

 Proof-of-concept: Antisense oligonucleotide mediated skipping of Fibrillin-1 exon 52

This is a highly interesting manuscript with great novelty, which provides new insights into the complex set of FBN1 mutations that all cause Marfan Syndrome. I still have a few suggestions.

  • It is true that there is very limited therapeutic options in Marfan Syndrome, however lately the use of angiotensin II receptor type 1 blockers (ARBs) is quite popular with Marfan patients and their physicians after a few reports on reduced aneurysm growth and aortic events. So the use of only beta-blockers is a bit outdated. This should be added in the introduction. Yet, if this is also symptoms management or actually interfering with the pathological process remains to be seen. (Circ Cardiovasc Genet. 2015 Apr;8(2):383-8. doi: 10.1161/CIRCGENETICS.114.000950. ; Lancet. 2019 Dec 21;394(10216):2263-2270. doi: 10.1016/S0140-6736(19)32518-8. ; Eur Heart J. 2020 Nov 14;41(43):4181-4187. doi: 10.1093/eurheartj/ehaa377. )

We thank the reviewer for this comment and this information has now been included in the manuscript

Line: 79-80

  • When searching the literature on Marfan fibroblasts it seems that many predicted dominant negative (DN) FBN1 mutations lead to impaired fibrillin-1 fiber formation, thus a haploinsufficient (HI) phenotype. It is important to actually measure fiber deposition because one cannot rely on predictions. This is shown here again and could be stressed more. The authors have shed light on a possible principle for this phenomenon, that the ratio between two different fibrillin-1 molecules 1:1 gives HI, while a ratio of 2:0 or 0:2 will rescue the imbalance.

Yet, many DN mutations are point mutations and thus to correct this, both the exon containing the mutation plus the wild type exon need to be deleted to rescue the phenotype probably, because two “misfolded” proteins may not rescue the fibers. I wonder if this would ever be ethical, to also change the normal FBN1 allele.

This reviewer has raised a most interesting point. Only further research in animal models can address this point and justify if there is a benefit after skipping exons in the transcripts from both alleles.

One further point is that the AO/PMO induced change in either allele is not permanent, ie this is not gene therapy where genome changes are introduced. This important point is now mentioned in the manuscript:

Line:  351-355

  • Alternatively, enhancing fibrillin-1 expression to generate more wild type fibrillin-1 may overcome the imbalance as demonstrated by Judge et al (J Clin Invest . 2004 Jul;114(2):172-81. doi: 10.1172/JCI20641.) where they add an additional wild type fbn1, which could rescue the Fbn1 C1039G/+ marfan mouse phenotype. In a way this is similar since here they change the ratio from 1:1 to 2:1. Even an alternative Fbn1 mutant (C1663R) could rescue the C1039G/+ phenotype partially. Moreover, HI Marfan patients with low expression of wild type FBN1 had a more severe aortic phenotype compared to patients with high FBN1 expression (Hum Mol Genet. 2015 May 15;24(10):2764-70. doi: 10.1093/hmg/ddv037.). While I think their technique indeed shows proof of concept, which is a very novel way of thinking and highly interesting, I think the applicability should be downsized a bit in the manuscript.

We thank the reviewer for these comments and would like to point out that this is a preliminary study that could be applicable to some FBN1 mutations, and future research is required to identify potentially redundant exons. There is much work to be done and the goal of this manuscript was to convey the novel approach to the wider scientific community.

Reviewer 3 Report

“Proof-of-concept: Antisense oligonucleotide mediated skipping of Fibrillin-1 exon 52” is a very interesting paper focused on the in vitro evaluation of antisense oligonucleotides designed to mediate the exclusion of an exon from FBN1. The result seems to lead to an internally truncated protein which is able to form fibres and it could represent a possible therapeutic approach for Marfan Syndrome.

All the experiments have been carried out in an exhaustive manner and only a few aspects must be improved:

  • Authors have to clarify how they have determined the exon skipping efficiency in terms of percentages (Fig.3b)
  • It could be interesting to face the future applications and perspectives of this antisense oligonucleotides technology for Marfan Syndrome.
  • Check figures and tables numbers, because an Error! makes difficult to find them.

Author Response

Reviewer 3

Comments and Suggestions for Authors

“Proof-of-concept: Antisense oligonucleotide mediated skipping of Fibrillin-1 exon 52” is a very interesting paper focused on the in vitro evaluation of antisense oligonucleotides designed to mediate the exclusion of an exon from FBN1. The result seems to lead to an internally truncated protein which is able to form fibres and it could represent a possible therapeutic approach for Marfan Syndrome.

All the experiments have been carried out in an exhaustive manner and only a few aspects must be improved:

  • Authors have to clarify how they have determined the exon skipping efficiency in terms of percentages (Fig.3b)

We have included additional information to the methods to clarify the calculations

Line: 491-495

Clarification was also added to the figure 3 caption

Line: 250

  • It could be interesting to face the future applications and perspectives of this antisense oligonucleotides technology for Marfan Syndrome.

  • Check figures and tables numbers, because an Error! makes difficult to find them.

This was apparently an editorial technical error that has been addressed

Reviewer 4 Report

In this report Lin Li. and colleagues reported the use of antisense oligonucleotides designed to mediate exclusion of FBN1 exon 52 during pre-mRNA splicing to restore monomer homology. They showed that exon 52 can be excluded in up to 100% of FBN1 transcripts in healthy control fibroblasts and verified by immunofluorescence assay a rescue of fibres with >80% skipping.

The study is very useful because they provide a concrete example of how induce targeted exon 52 skipping from full-length FBN1 mRNA transcripts expressed in healthy control fibroblasts by using antisense oligonucleotides. This methodology may be an attractive therapeutic option for the treatment of Marfan syndrome but can be included as a therapeutic approach in other human diseases.   

It is my opinion that in this form this work does not present a substantial evaluation concerning the efficiency and the effect of PMO52 that makes it suitable for publication at this stage. Major, comprehensive, and meticulous revisions are needed before publication.

  1. In the immunofluorescence study the authors detected a fibrillin-1 protein in order to demonstrated a reduction of fibers in treated control cell lines. Also in control and in patient lines, treatment with PM052 resulted in the rescue of fibrillin-1 fibers. In order to evaluate the alteration of FBN1protein level it is necessary apply a quantification study of relative FBN1-intensity signal by analyzing in three independent experiments an appropriate cell numbers. The only representative immunofluorescence imagines are not sufficient in order to assess the rescue of FBN1 protein.
  2. Also, to use fibroblast patient cell lines I suggest to evaluate the protein level (by western blotting) or mRNA level (by Q-PCR) of FBN1 and the corresponding rescue in patient cell lines.
  3. In order to verify the efficacy of the proposed methodology, we suggest to validate the functional activity of FBN1 protein by additional assays by monitoring the level and the activity of one (or more) marker that are associated to FBN1 signaling. (ei. monitoring the level of TGFB-target protein as ERK1/2, p-Smad2, Smad3….). I believe that another validation experiment could be necessary in order to test the efficacy of the methodology.

Minor REVISIONS:

  1. Results: report the RefSeq of FBN1 transcript (NM_XXX).
  2. ESE and ISE what does mean? Explain the definition of the acronymous
  3. Methods: report the number of cells observed in the microscopy study.
  4. Report exactly the figure number in the text.

Author Response

Reviewer 4

Comments and Suggestions for Authors

In this report Lin Li. and colleagues reported the use of antisense oligonucleotides designed to mediate exclusion of FBN1 exon 52 during pre-mRNA splicing to restore monomer homology. They showed that exon 52 can be excluded in up to 100% of FBN1 transcripts in healthy control fibroblasts and verified by immunofluorescence assay a rescue of fibres with >80% skipping.

We are not sure who Lin Li is. The first author is Jessica Cale.

The study is very useful because they provide a concrete example of how induce targeted exon 52 skipping from full-length FBN1 mRNA transcripts expressed in healthy control fibroblasts by using antisense oligonucleotides. This methodology may be an attractive therapeutic option for the treatment of Marfan syndrome but can be included as a therapeutic approach in other human diseases.   

It is my opinion that in this form this work does not present a substantial evaluation concerning the efficiency and the effect of PMO52 that makes it suitable for publication at this stage. Major, comprehensive, and meticulous revisions are needed before publication.

As we have mentioned in the manuscript, this work is preliminary and will require intense follow-up studies, including animal modelling.

  • In the immunofluorescence study the authors detected a fibrillin-1 protein in order to demonstrated a reduction of fibers in treated control cell lines. Also in control and in patient lines, treatment with PM052 resulted in the rescue of fibrillin-1 fibers. In order to evaluate the alteration of FBN1protein level it is necessary apply a quantification study of relative FBN1-intensity signal by analyzing in three independent experiments an appropriate cell numbers. The only representative immunofluorescence imagines are not sufficient in order to assess the rescue of FBN1 protein.

As mentioned previously, biological replicates were carried out (n=3) and one set has been now included in the supplementary data.

Line: 282-284

In addition, we are uncomfortable with the word “rescue” with respect to re-appearance of fibre-like fibrillin 1 staining and have replaced this with “re-appearance” or “formation” in the manuscript. We feel that rescue implies functional improvements/changes that we have not yet investigated. Clearly the re-appearance of fibrillin 1 staining in the extracellular matrix of MFS patient cells is a step forward but the function has to be validated.

Line: 22, 275, 333

  • Also, to use fibroblast patient cell lines I suggest to evaluate the protein level (by western blotting) or mRNA level (by Q-PCR) of FBN1 and the corresponding rescue in patient cell lines.

The RT-PCR was carried out with 20 cycles of amplification, well within semi-quantitative range. Since amplicon signals from normal, patient, treated and untreated cells were similar, we may assume that the mRNA levels have not changed substantially. Since the skipping of exon 52 does not disrupt the reading frame, nonsense mediated decay of the mRNA should not be a factor.

We did attempt western blotting from conditioned media and cells (healthy and patient) but were unable to generate reliable and consistent signals.

  • In order to verify the efficacy of the proposed methodology, we suggest to validate the functional activity of FBN1 protein by additional assays by monitoring the level and the activity of one (or more) marker that are associated to FBN1 signaling. (ei. monitoring the level of TGFB-target protein as ERK1/2, p-Smad2, Smad3….). I believe that another validation experiment could be necessary in order to test the efficacy of the methodology.

The reviewer raises very valid points, but these were not in the scope of this paper. This work will be undertaken and be the subject of future papers.

Minor REVISIONS:

  • Results: report the RefSeq of FBN1 transcript (NM_XXX).

The RefSeq number has been added to the results and methods sections

Line: 139, 449-451

  • ESE and ISE what does mean? Explain the definition of the acronymous

Explanation of the exon splice enhancer (ESE) and intronic splice enhancer(ISE) acronyms have now been added to the manuscript.

Line: 145-146

  • Methods: report the number of cells observed in the microscopy study.

One hundred thousand cells were plated onto coverslips for the microscopy study.

  • Report exactly the figure number in the text.

The figure numbers have been addressed.

Round 2

Reviewer 1 Report

In general, all changes made by the authors were text changes. The important experiments and controls required to improve this manuscript were not done.  

Detailed comments:

  1. Even if the authors changed the description of some results, the quality of the results was not improved and remains not convincing for several aspects, without proper quantification and biological replicates. There are established methods for quantification available in the literature. Biological replicates are different from technical replicates, the authors do not seem to understand this aspect.
  2. Absolutely no efforts to try to experimentally address the critical comments.
  3. The RT-PCR results are still not convincing due to the lack of expression evidence. It was not justified why a 20-cycle is “well within semi-quantitative”.
  4. The authors seem not to understand why it is absolutely critical to perform transfection controls in order to be able to draw any conclusions from the experiments. The authors state “we are confident that….”. This is not how quality science is done, confidence comes with the proper controls.
  5. It is still not meaningful to propose personalized medicine on specific mutants on MFS, even if there are 8 patients. How should a clinical trial ever be conducted on 8 patients? The authors should not ignore this important fact if they want to discuss and propose personalized medicine, even it is a proof-of-concept.

Author Response

In general, all changes made by the authors were text changes. The important experiments and controls required to improve this manuscript were not done.  

Detailed comments:

  1. Even if the authors changed the description of some results, the quality of the results was not improved and remains not convincing for several aspects, without proper quantification and biological replicates. There are established methods for quantification available in the literature. Biological replicates are different from technical replicates, the authors do not seem to understand this aspect.

Biological replicates are challenging  since cells derived from only one Marfan patient were available for this study, and therefore, in addition to performing the transfections multiple times (technical replicates n=3), we transfected cells at different passages, with reproducible outcomes.

  1. Absolutely no efforts to try to experimentally address the critical comments.

Please see comments below

  1. The RT-PCR results are still not convincing due to the lack of expression evidence. It was not justified why a 20-cycle is “well within semi-quantitative”.

            We did not intend to confer highly specific quantitation of FBN1 mRNA levels before and after treatment in these studies, but indicated that overall levels were similar before and after treatment with splice switching or control AOs. The semi-quantitative results presented are a comparison of the full-length and exon skipped products within a single sample with no comparison made between samples. We were fortunate in this case that FBN1 mRNA levels are substantial and could be detected with 20 cycles (or less). As the reviewer is likely aware from studies decades ago that 20 cycles is in the early log-linear phase of DNA amplification.

From Wong and Medrano (Marisa L. Wong and Juan F. Medrano USA BioTechniques 39:75-85 (July 2005)

  1. The authors seem not to understand why it is absolutely critical to perform transfection controls in order to be able to draw any conclusions from the experiments. The authors state “we are confident that….”. This is not how quality science is done, confidence comes with the proper controls.

We are unsure what the reviewer means by transfection controls in the experiments. We have designed many different AO sequences targeting regions of the FBN1 pre-mRNA, with two different chemistries (first as 2Omethyl modified bases on a PS backbone and then using the morpholino chemistry). These compounds were transfected under conditions optimized for the cells (cationic liposomes for the 2OMe-PS AOs and nucleofection for the morpholinos). As anticipated, some of the initial 2OMe AOs were effective, that is induced specific exon skipping and some did not. Since we have an assay that unequivocally demonstrates AO effect (appearance of an altered transcript product) we could ascertain the more promising compounds for further experimentation using the morpholino chemistry. In all experiments, we used negative transfection controls (unrelated sequences) to show the oligo chemistry was not involved in changing specific FBN1 expression. It is not practical to use random control sequences for each active AO as so many different compounds are evaluated during the screening process. The morpholino negative control is a sequence provided by GeneTools as a standard control and is commercially available. Since the goal of this work is to develop effective oligos that excise a specific exon from the target transcript, effective (positive) oligos are identified and those compounds found to be entirely ineffective can be repurposed as additional negative controls.

The only other “transfection controls” we could have included would have been fluorescently tagged oligos to show cellular/nuclear uptake (similar work was reported decades ago) but this does not indicate biological activity and hence was not included in this manuscript.

  1. It is still not meaningful to propose personalized medicine on specific mutants on MFS, even if there are 8 patients. How should a clinical trial ever be conducted on 8 patients? The authors should not ignore this important fact if they want to discuss and propose personalized medicine, even it is a proof-of-concept.

We appreciate the reviewers hesitance for small scale clinical trials, however the reviewer does not seem to be aware of the fact that a meaningful clinical trial, using a splice switching oligo targeting a CLN7 mutation was carried out on ONE patient and resulted in FDA approval: Milasen (https://cen.acs.org/business/Milasen-drug-idea-injection-10/97/i42). We did not include this in the manuscript as the aim of this work was to show proof-of-concept that the exon skipping strategy described in the current study could result in FBN1 expression. We would also like to note that we hypothesise that this exon skipping strategy will be applicable more broadly than to only the single mutation presented in this manuscript. In the discussion we mention and acknowledge that the functionality of the new FBN1 isoform (Δ52) is yet to be determined, but its re-appearance is a promising first step. In many respects, this work is similar to our early in vitro work in the mdx mouse, a dystrophinopathy model, where we showed re-appearance of the missing protein (after skipping an exon carrying a nonsense mutation) but could not assess functionality. Here, we induce a mutation in the normal allele that results in the appearance of extruded FBN1, albeit of undetermined function but presumably better than none at all.

Reviewer 4 Report

About comment number 2 (Also, to use fibroblast patient cell lines I suggest to evaluate the protein level (by western blotting) or mRNA level (by Q-PCR) of FBN1 and the corresponding rescue in patient cell lines.) I suggest to report in the text the importance of protein level detection justifying the technical problems that emerged from western blot essays.

Author Response

About comment number 2 (Also, to use fibroblast patient cell lines I suggest to evaluate the protein level (by western blotting) or mRNA level (by Q-PCR) of FBN1 and the corresponding rescue in patient cell lines.) I suggest to report in the text the importance of protein level detection justifying the technical problems that emerged from western blot essays.

We thank the reviewer for the suggestion and hope we have now addresses this adequately. The following sentences were added to the discussion section to better explain why western blotting results were not included. “Western blotting analysis of intracellular and extracellular fibrillin-1 was attempted, however, due to poor signal and resolution in samples from healthy control fibroblasts we were unable to confirm any changes in fibrillin-1 abundance. Further optimisation of the western blot protocol to produce reliable results is essential to assess the effect of FBN1 exon skipping on the abundance of fibrillin-1.” Line: 428-433

Round 3

Reviewer 1 Report

The authors have made changes in the text to further clarify several aspects. However, some important conclusions remain not justified due to the lack of additional experimental data (previously requested). Also, some previous comments were simply ignored. There also may be some misunderstanding what exactly is needed. Here are the aspects that still needs experimental attention:

  1. The authors argued that biologicals replicates for MFS are challenging since cells derived from only one Marfan patient were available for this study. While it is clear that only cells from one MFS patient are available, the authors did not respond to the previous request to include “biological replicates of normal controls” (first review). These are commercially available. This is important, because fibrillin-1 expression, assembly and deposition can vary significantly between primary cells from different donors.

  1. Proper quantifications of the intensities of fibers are still not provided, despite the fact that feasible quantification methods have been published. This is important given the low signal/noise ratio of the fibrillin staining apparent in the Suppl. Fig. S3, the few fibers in the treated sample relative to the untreated control, and the bias picking zoomed-in regions of the PMO52 treated samples in Figs. 3c and 4b (compared to Fig. S3). A sufficient number of images in overview, as shown in S3, should be quantified and the data should be included.

  2. In the 1st review, this reviewer requested qPCR and Western blotting to determine the expression levels of fibrillin-1 mRNA and protein, as complementing experiments for Fig. 3c and 4b. This is critical, to conclude that the few fibers in the cell cultures treated with 50 uM PMO52 (resulting ~50% exon skipping) is indeed due to dominant negative effects, and not simply due to reduced fibrillin-1 mRNA and protein expression.
    On the other side, the semi-quantitative results shown in Fig 3a and 4a are convincing.

  3. For Fig. 2, the authors intended to choose the sequence that shows highest exon skipping for following experiments. The authors also acknowledge the feasibility of transfection control in their rebuttal. Since the efficiency of transfection may differ, the current conclusions that AO52.1n is the most efficient is simply not justified. If the purpose was just to identify one or more AOs that work, this would be a different approach which does not necessarily require a transfection control. But this is not how the data are presented.

Author Response

Major revisions

Comments:

The authors have made changes in the text to further clarify several aspects. However, some important conclusions remain not justified due to the lack of additional experimental data (previously requested). Also, some previous comments were simply ignored. There also may be some misunderstanding what exactly is needed. Here are the aspects that still needs experimental attention:

  1. The authors argued that biologicals replicates for MFS are challenging since cells derived from only one Marfan patient were available for this study. While it is clear that only cells from one MFS patient are available, the authors did not respond to the previous request to include “biological replicates of normal controls” (first review). These are commercially available. This is important, because fibrillin-1 expression, assembly and deposition can vary significantly between primary cells from different donors.

We are aware that fibrillin-1 expression can differ between individuals and have evaluated exon skipping, and separately fibrillin-1 immunofluorescence in unrelated untreated healthy control cell lines. When assessing exon skipping at the RNA level in two healthy control fibroblast lines no significant differences in exon skipping efficiency were observed over a range of transfection conditions. The levels of exon skipping, as assessed by semi-quantitative RT-PCR was similar in the unrelated healthy cells.

However, when undertaking protein studies after transfection we encountered poor adherence of our second healthy cell line to coverslips and were unable to complete immunofluorescence studies of fibrillin-1 in that cell line. However, we would like to point out that we are not directly comparing the fibres in the heathy control to the patient, rather using it as a baseline of what fibres can look like in the healthy control. We demonstrate in the manuscript that a complete lack of fibres, after inducing ~50% FBN1 exon skipping to mimic the splice mutation, in healthy cells is not the ‘normal’ state and reflects the disease condition in vitro. Further transfection of healthy controls cannot be undertaken in the requested timeframe as the first author has completed her PhD candidature and left the laboratory. Transfection and induced FBN1 exon skipping in normal cells will be undertaken when another student resumes the fibrillin exon skipping studies and the consequences of skipping other exons are evaluated.  We will ensure that multiple unrelated healthy biological replicates are included.

  1. Proper quantifications of the intensities of fibers are still not provided, despite the fact that feasible quantification methods have been published. This is important given the low signal/noise ratio of the fibrillin staining apparent in the Suppl. Fig. S3, the few fibers in the treated sample relative to the untreated control, and the bias picking zoomed-in regions of the PMO52 treated samples in Figs. 3c and 4b (compared to Fig. S3). A sufficient number of images in overview, as shown in S3, should be quantified and the data should be included.

The reviewer refers to published quantification methods, however despite our previous response stating that we have been unable to find such quantification methods, we are unsure how to go about this request. From our previous studies on exon skipping and dystrophin expression, we are very conscious of the limitations of accurate quantification by immunofluorescence.  We used this technique to only indicate presence or absence of the fibrillin fibres. Simple relative-intensity quantification will not distinguish if the fibrillin-1 monomers are assembling into fibres, intracellular or diffuse extracellular, and therefore is not appropriate for the current study. One paper (Godfrey et al, 1995, Am J Pathol, 146:1414-1421) provides ‘quantification’ based on the fragmented morphology of fibrillin-1 fibres, and it is from this paper that we used the term ‘fragmented’ in our manuscript. As we previously stated, accurately measuring the length of fibres was impossible due to their lattice or mesh-like deposition.

  1. a) In the 1st review, this reviewer requested qPCR and Western blotting to determine the expression levels of fibrillin-1 mRNA and protein, as complementing experiments for Fig. 3c and 4b. This is critical, to conclude that the few fibers in the cell cultures treated with 50 uM PMO52 (resulting ~50% exon skipping) is indeed due to dominant negative effects, and not simply due to reduced fibrillin-1 mRNA and protein expression.

We have discussed and added to the manuscript, that we attempted and were unable to generate reliable western blotting data. Due to the relatively high abundance of FBN1 mRNA and the ability to undertake the RT-PCR with only 20 cycles (early log exponential stage of amplification), we did not see any substantial changes in mRNA levels between these samples.  Consequently, we submit that the FBN1 mRNA levels are not reduced, consistent with the findings of Liu, et al. (Multi-Exon Deletions of the FBN1 Gene in Marfan Syndrome. BMC Med. Genet. 2001, 2, 11.). The exon skipping mutation described by Liu et al was also reported to result in decreased fibrillin-1 synthesis, as we state this in the manuscript and speculate that this is one of the reasons for the lower fibre abundance observed. We must emphasize that the exon skipping strategy proposed here can never be regarded as a “cure”. Rather, targeted FBN1 exon skipping of some mutations could allow abnormal proteins to pair and assemble in fibrils. These proteins are obviously not normal, and this could also contribute to reduced protein expression. Nevertheless some residual function could confer benefits compared to no FBN1 expression (as seen by immunofluorescence).  This situation is exactly what we encountered with our early dystrophin exon skipping work as a treatment for Duchenne muscular dystrophy.  We could treat cells (human or animal) carrying a null mutation in the dystrophin gene and see the re-appearance of a dystrophin isoform by immunofluoresence, Then, as now there was no way to assess function of that isoform.

  1. b) On the other side, the semi-quantitative results shown in Fig 3a and 4a are convincing.

Thank you

  1. For Fig. 2, the authors intended to choose the sequence that shows highest exon skipping for following experiments. The authors also acknowledge the feasibility of transfection control in their rebuttal. Since the efficiency of transfection may differ, the current conclusions that AO52.1n is the most efficient is simply not justified. If the purpose was just to identify one or more AOs that work, this would be a different approach which does not necessarily require a transfection control. But this is not how the data are presented.

We are uncertain as to the reviewer’s comment/question here or if this is semantics.  If the reviewer is stating that we cannot say AO52 is the most efficient, we would certainly agree. We have screened a number of AOs, and compared these under parallel transfection conditions to identify those compounds that induced more robust exon skipping than others in titration experiments, as we have outlined in the manuscript.

Nevertheless, there is always room for further improvement in AO design.  Very subtle changes in AO design, microwalking, shifting the annealing coordinates by one base and/or then reducing the length of the AO by a single base could easily allow for another 20+ compounds to be evaluated.  Even then, the choice of a “pre-clinical candidate” could be influenced by the chemistry of the AO and synthesis efficiencies, but the gist of this manuscript was to convey the concept and set the scene for future animal model validation. 

For the purposes of this study, we sought to develop a compound that induced robust FBN1 exon skipping and then begin to study the consequences.
